# Endothelial Jagged1 Antagonizes Dll4/Notch Signaling in Decidual Angiogenesis during Early Mouse Pregnancy

**DOI:** 10.3390/ijms21186477

**Published:** 2020-09-05

**Authors:** Nicole M. Marchetto, Salma Begum, Tracy Wu, Valerie O’Besso, Christina C. Yarborough, Nuriban Valero-Pacheco, Aimee M. Beaulieu, Jan K. Kitajewski, Carrie J. Shawber, Nataki C. Douglas

**Affiliations:** 1Department of Obstetrics, Gynecology and Reproductive Health, Rutgers New Jersey Medical School, Newark, NJ 07103, USA; nm949@njms.rutgers.edu (N.M.M.); sb1802@njms.rutgers.edu (S.B.); tw472@njms.rutgers.edu (T.W.); 2Rutgers New Jersey Medical School, Newark, NJ 07103, USA; valerie.obesso@gmail.com; 3School of Graduate Studies, Rutgers Biomedical and Health Sciences, Newark, NJ 07103, USA; ccy21@gsbs.rutgers.edu; 4Department of Microbiology, Biochemistry, and Molecular Genetics, Rutgers Biomedical and Health Sciences, Newark, NJ 07103, USA; pv164@njms.rutgers.edu (N.V.-P.); ab1550@njms.rutgers.edu (A.M.B.); 5Center for Immunity and Inflammation, Rutgers Biomedical and Health Sciences, Newark, NJ 07103, USA; 6Department of Physiology & Biophysics, University of Illinois Chicago, Chicago, IL 60612, USA; kitaj@uic.edu; 7Department of Obstetrics and Gynecology, Division of Reproductive Sciences, Columbia University College of Physicians and Surgeons, New York, NY 10032, USA; cjs2002@cumc.columbia.edu

**Keywords:** Notch, Jag1, Dll4, endothelial cells, decidua, angiogenesis, capillaries, spiral arteries

## Abstract

Maternal spiral arteries and newly formed decidual capillaries support embryonic development prior to placentation. Previous studies demonstrated that Notch signaling is active in endothelial cells of both decidual capillaries and spiral arteries, however the role of Notch signaling in physiologic decidual angiogenesis and maintenance of the decidual vasculature in early mouse pregnancy has not yet been fully elucidated. We used the *Cdh5-Cre^ERT2^;*
*Jagged1(Jag1)^flox/flox^* (*Jag1∆EC*) mouse model to delete Notch ligand, *Jag1*, in maternal endothelial cells during post-implantation, pre-placentation mouse pregnancy. Loss of endothelial *Jag1* leads to increased expression of Notch effectors, *Hey2* and *Nrarp,* and increased endothelial Notch signaling activity in areas of the decidua with remodeling angiogenesis. This correlated with an increase in Dll4 expression in capillary endothelial cells, but not spiral artery endothelial cells. Consistent with increased Dll4/Notch signaling, we observed decreased VEGFR2 expression and endothelial cell proliferation in angiogenic decidual capillaries. Despite aberrant Dll4 expression and Notch activation in *Jag1∆EC* mutants, pregnancies were maintained and the decidual vasculature was not altered up to embryonic day 7.5. Thus, Jag1 functions in the newly formed decidual capillaries as an antagonist of endothelial Dll4/Notch signaling during angiogenesis, but Jag1 signaling is not necessary for early uterine angiogenesis.

## 1. Introduction

During early pregnancy, from embryo implantation to placentation, formation of new capillaries and physiologic remodeling of uterine spiral arteries (SpAs), which are distal branches of the maternal uterine arteries, are essential for normal embryonic growth and development [1,2,3]. Abnormal uterine vascular development early in pregnancy can set off a “ripple effect”, resulting in poor placentation, reduced placental function and adverse downstream effects, such as abnormal growth and development of the embryo and pregnancy failure by mid-gestation [1,4]. Thus, understanding the pathways that regulate pre-placentation vascular development in the uterus will help to better elucidate the pathways that impact maintenance and success of pregnancy at later stages.

In mice, uterine vascular changes are initiated by embryo implantation. Embryo implantation, which occurs by embryonic day (E) 4.5, triggers uterine stromal cell decidualization and concomitant decidual angiogenesis [1,2,5,6,7,8]. Blood vessels grow into the uterus on the mesometrial side creating a mesometrial-anti-mesometrial polarity of the implantation site and dividing the implantation site into two regions, mesometrial region and anti-mesometrial region [1,7] (Figure 1A–C). The mesometrial region can be further divided into the mesometrial pole and central region. In the central region of the mesometrial decidua, sprouting angiogenesis is most pronounced at E4.5 and remodeling angiogenesis occurs from E6.5–E8.5 [7]. In the anti-mesometrial region, angiogenesis begins at E4.5 and initiates endothelial cell (EC) proliferation, which peaks by E6.5, and results in a greater than 2-fold increase in vascular density at E8.5, as compared to E4.5 [7]. From E4.5–E7.5, decidual angiogenesis creates a rich, new capillary network that supports the growing pregnancy prior to placentation.

Placenta formation begins at E8.5 and is complete by mid-gestation, or E10.5 [1,2]. Transformation of the maternal SpAs, initiates at E6.5 and is the key vascular change occurring from E8.5–E10.5. Embryo-derived trophoblasts migrate from the implantation site to the SpAs and mediate SpA remodeling [3,9,10]. SpAs, like most arteries, are comprised of two key cell types, ECs, which make up the inner lining of the vessel, and mural cells, which include pericytes and vascular smooth muscle cells, that surround the endothelium and are essential for maintenance of vascular integrity [11,12,13]. During the process of SpA remodeling, trophoblasts intercalate the vessels, replacing both ECs and vascular smooth muscle cells, and transform constricted artery-like structures into dilated and enlarged vein-like SpAs, which will support adequate blood flow through to the placenta [1,3,14,15]. Prior to mid-gestation, E10.5, two distinct vascular events, decidual angiogenesis and SpA remodeling, are essential for normal placentation and placental function.

The Vascular Endothelial Growth Factor (VEGF) and Notch signaling pathways are interconnected and together have been shown to be critical for vascular development of the retina, embryo and ovary [16,17,18,19]. VEGF signaling is essential for vascular proliferation and expansion in decidual angiogenesis and is necessary to support early pregnancy [5,7]. Experimental mouse models with targeted deletions of Notch signaling pathway members possess a variety of vascular defects that cause embryonic lethality by mid-gestation [20,21,22,23,24,25,26], but loss of Notch signaling in decidual angiogenesis and vascular remodeling in early pregnancy has not been fully elucidated.

Notch proteins are expressed in ECs and mural cells. Vascular ECs express receptors, Notch1 and Notch4, and ligands delta-like (Dll) 1, Dll4, Jagged1 (Jag1) and Jag2 [23,27,28,29,30,31]. Mural cells have been previously shown to express Notch1, Notch3 and Jag1 [26,27,32,33,34]. Ligand binding of Notch triggers release of the Notch intracellular domain (NICD), which translocates into the nucleus and promotes transcription of downstream effectors of Notch signaling, including members of the Hairy/Enhancer of Split related with YRPW motif (*Hey*) families and Notch-regulated ankyrin repeat-containing protein (*Nrarp*) [29,35,36]. Notch activation can be elicited by ligands expressed within the same cell (*cis*-activation) as the Notch receptor or on an adjacent cell (*trans*-activation) [37]. Alternatively, blockade of an inhibitory *cis*-interaction between receptor and ligand can drive Notch activation [38].

We have previously shown that Dll4 is expressed in the ECs of SpAs, while Jag1 is expressed in both ECs and pericytes of SpAs [27]. Consistent with ligand expression, Notch signaling is active in ECs in decidual capillaries and SpAs [27,39]. Herein, we investigated the role of endothelial Jag1/Notch signaling in the formation and maintenance of the maternal decidual vasculature in early mouse pregnancy. Mice carrying an EC-specific, tamoxifen inducible Cre recombinase, *Cdh5-Cre^ERT2^* [40] and a *Jagged1^flox/flox^* alleles [41] were used to achieve cell-type specific deletion of *Jag1* during decidual angiogenesis. In order to assess the decidual vasculature after embryo implantation and prior to placentation, we evaluated pregnancies at E7.5, a stage of pregnancy when decidual angiogenesis is nearing completion and decidual vascular density is greatest.

## 2. Results

### 2.1. Characterization of Vasculature in the Mouse Implantation Site at E7.5

To characterize the vasculature in the peri-implantation uterus, we immunostained wild-type implantation sites with EC marker, CD31, and mural cell markers, platelet-derived growth factor receptor beta (PDGFRβ), neural/glial antigen 2 (NG2), and α-smooth muscle actin (SMA) [12,42,43,44]. Visualization of the whole implantation site demonstrates that NG2 expression is enriched around the SpAs at the mesometrial pole of the mesometrial region while it is variably expressed in decidual vessels in the central region and anti-mesometrial region (Figure 1C). At the mesometrial pole (Figure 1D), mural cells expressing NG2, PDGFRβ and SMA are adjacent to but not overlapping with CD31^+^ ECs, consistent with these vessels being the SpAs. In contrast, capillaries at the mesometrial pole of the mesometrial region are comprised of CD31^+^ ECs that lack SMA^+^ mural cell coverage. In the central region of the mesometrial region, the expression of NG2 is sparse (Figure 1E). In the anti-mesometrial region, NG2^+^ and PDGFRβ^+^ cells are closely associated with CD31^+^ ECs, while co-expression is not noted (Figure 1F). SMA expression was not detected in the central region or anti-mesometrial region. The morphology and pattern of expression of the NG2^+^ and PDGFRβ^+^ cells in the anti-mesometrial region, and lack of SMA expression suggest that these mural cells are pericytes. Together, these data show regional differences in the vasculature at E7.5; mature, non-angiogenic resident SpAs with their associated mural cells are present in the mesometrial decidua, whereas angiogenic capillary networks make up the central and anti-mesometrial regions of the decidua.

### 2.2. Characterization of Notch Expression in the Decidual Vasculature

To determine expression of Notch proteins in the decidual vasculature, implantation sites were double stained with antibodies to detect Notch1 and Notch4 with respect to ECs and mural cells. In the mesometrial region, CD31^+^ capillary ECs at the mesometrial pole and in the central region express Notch1 and Notch4 (Figure 2A,B). In SpAs at the mesometrial pole, PDGFRβ^+^ mural cells are adjacent to ECs that express both Notch1 and Notch4 (Figure 2A). In the anti-mesometrial region, Notch1 is expressed in CD31^+^ ECs surrounded by Notch1 negative PDGFRβ^+^ mural cells (Figure 2C). Notch4 expression was not detected in the anti-mesometrial region (data not shown).

Double staining for Jag1 or Dll4 and a vascular cell marker revealed that these Notch ligands are dynamically expressed in the decidual vasculature. At the mesometrial pole of the mesometrial region, CD31^+^ ECs in capillaries and SpAs express both Jag1 and Dll4 (Figure 3A,D). A subset of the NG2^+^ and PDGFRβ^+^ mural cells in the SpAs also express Jag1, but not Dll4 (Figure 3A,D). In the central region of the mesometrial region, expression of Jag1 is not detected (Figure 3B), while Dll4 is expressed in a perinuclear, non-vascular pattern in decidual stromal cells (Figure 3E). In the anti-mesometrial region, Dll4 is expressed in CD31^+^ ECs and Dll4^+^ cells are associated with PDGFRβ^+^ mural cells. Expression of Jag1 is scant in the anti-mesometrial region but is found in CD31^+^ ECs and surrounding PDGFRβ^+^ mural cells. Together, these data demonstrate that capillary ECs in the mesometrial region and anti-mesometrial region and SpA ECs co-express Notch1, as well as Jag1 and Dll4, suggesting these proteins mediate Notch signaling to regulate decidual angiogenesis and SpA remodeling.

### 2.3. EC-Specific Deletion of Jag1 in Early Pregnancy

To determine the role of endothelial Jag1 in the decidual vasculature in early pregnancy, we used a tamoxifen inducible driver, *Cdh5-Cre^ERT2^*, to express Cre recombinase in the endothelium and delete *Jag1* during decidual angiogenesis. Tamoxifen was administered at embryo implantation, E4.5, and pregnancies and uterine phenotypes were assessed at E7.5, the end of the active period of decidual angiogenesis (Figure 4A). Notch proteins and ligands are expressed in the ovarian endothelium, including in ECs of corpora lutea [45] (Appendix A). Given the requirement for progesterone secretion from ovarian corpora lutea to maintain early pregnancy, progesterone supplementation was initiated at E4.5 to overcome potential defects in *Jag1∆EC* mutant ovaries. To determine the recombination efficiency of the *Cdh5-Cre^ERT2^* line using this tamoxifen regimen, *Cdh5-Cre^ERT2^* mice were crossed to the *ROSA26 tdTomato* reporter strain (Figure 4B). Expression of fluorescent tdTomato protein throughout the implantation site is observed in a pattern similar to CD31 in the mesometrial region and anti-mesometrial region (Figure 1C). Implantation sites were stained with endothelial cell marker, CD31, or mural cell marker, NG2, to assess tdTomato expression with respect to ECs and mural cells. Co-expression is observed with CD31^+^ ECs, but not with NG2^+^ mural cells (Appendix A). tdTomato is expressed in 94–98% of CD31^+^ ECs in the mesometrial and anti-mesometrial regions and in 90% of NG2^+^ SpAs in the mesometrial region (Appendix A). These data demonstrate efficient recombination in decidual ECs of capillaries and SpAs at E7.5.

To evaluate the role of endothelial Jag1 in post-implantation, pre-placentation pregnancy, uteri from pregnant *Jag1∆EC* and *Cdh5-Cre^ERT^*^2^ control females were isolated at E7.5. A decrease in Jag1 is observed in the decidua ECs of *Jag1ΔEC* uteri as compared to controls (Figure 4C,D). In contrast, analysis of the uteri from *Jag1**ΔEC* pregnancies demonstrates similar expression of Jag1 in embryos of control and *Jag1**ΔEC* mutants. qRT-PCR performed on whole implantation sites shows a significant decrease in *Jag1* expression in *Jag1**ΔEC* mutants as compared to controls (Figure 4E). To quantify Jag1 expression in ECs and SpAs at the mesometrial pole, implantation sites were stained to detect Jag1 in CD31^+^ ECs and Jag1 expressed in NG2^+^ SpAs (Figure 4F,G). Expression of Jag1 is significantly lower in all CD31^+^ ECs (Figure 4F,G) in *Jag1ΔEC* pregnancies when compared to controls, demonstrating similar efficacy of *Cdh5-Cre^ERT2^-*mediated deletion of *Jag1* in capillary and SpA ECs, as seen for the *ROSA26 tdTomato* reporter, and confirming that our model systems works.

### 2.4. Loss of Endothelial Jag1 Does Not Impact Pregnancy at E7.5

To determine if EC-specific loss of *Jag1* impacts pregnancy and embryo development at E7.5, implantation sites were collected and counted, and embryo morphology was assessed in H&E stained sections. Litter size for *Cdh5-Cre^ERT2^* control and *Jag1**ΔEC* pregnancies is similar (Figure 5A). Embryo development involves the formation and disappearance of morphologic structures during gestation [46]. To determine embryo morphology with respect to embryonic age, all tissue sections containing embryos were scored for the presence or absence of the following morphologic structures: primitive streak, allantois, cranial neural fold and somites (Figure 5B,C). These morphologic “landmarks” are common to gastrulating wild-type embryos between E6.5 and E8.5, and the timing of their appearance is used to compare stage of embryo development. Comparison of *Cdh5-Cre^ERT2^* control and *Jag1**ΔEC* pregnancies revealed that embryos from *Cdh5-Cre^ERT2^* control and *Jag1**ΔEC* pregnancies progressed to a similar stage of development (Figure 5D).

### 2.5. Expression of Notch Ligand, Dll4, and Notch Effectors Is Increased in Jag1∆EC Pregnancies at E7.5

To assess the impact of loss of endothelial *Jag1* on expression of Dll4, *Jag1**ΔEC* and *Cdh5-Cre^ERT2^* uteri were double stained for CD31 or NG2, and Dll4. The percentage of CD31^+^ ECs expressing Dll4 was quantified in the mesometrial and anti-mesometrial regions. At the mesometrial pole and in the anti-mesometrial region, Dll4 expression is significantly increased in the CD31^+^ capillary ECs in *Jag1∆EC* pregnancies compared to control pregnancies (Figure 6A,B). Dll4 expression is increased but not significantly, in the ECs of the NG2^+^ cell covered SpAs (Figure 6C). These data show that loss of *Jag1* in ECs leads to increased EC Dll4 expression. To understand the impact of EC-specific loss of *Jag1* and increased EC Dll4 expression on Notch signaling, we assessed the expression of downstream effectors of Notch signaling. Total RNA was isolated from *Jag1**ΔEC* mutant and control whole implantation sites, from which myometrium was removed. Expression of direct Notch target genes was determined by qRT-PCR. Expression of Notch effectors, *Hey2* and *Nrarp*, is significantly increased in *Jag1∆EC* mutants relative to control pregnancies. These data suggest increased Dll4/Notch signaling in the implantation sites of *Jag1∆EC* mutants (Figure 6D).

### 2.6. Jag1/Notch Signaling Regulates Angiogenic Gene Expression and Endothelial Proliferation in the Anti-Mesometrial Decidua

Dll4/Notch signaling has been shown in the retina to suppress VEGFR2 expression leading to reduced EC proliferation [17]. To evaluate the impact of EC-specific loss of *Jag1* on endothelial Dll4/Notch signaling, we determined the expression of Notch1 with an antibody against the cytoplasmic domain, the Notch1 ICD (N1ICD). We also determined expression of VEGFR2 and EC proliferation. *Jag1* inactivation increases nuclear expression of N1ICD in anti-mesometrial capillary ECs. This increase in nuclear Notch1 expression is consistent with an increase in expression of Notch1 protein and in EC Notch1 signaling in *Jag1∆EC* mutants (Figure 7A). We found N1ICD expression in cells adjacent to Dll4^+^ cells in the capillaries of *Jag1∆EC* mutants, suggesting *trans*-activation of Notch1 by Dll4 (Figure 7B). The percentage of CD31^+^ ECs in the anti-mesometrial decidua expressing Ki67 (Figure 7C) and VEGFR2 (Figure 7D) is significantly decreased in *Jag1∆EC* mutants relative to controls. In contrast, expression of VEGFR2 in the myometrium (Figure 7D, black arrowheads) is similar in *Jag1∆EC* mutants and controls. Together, the data show that increased Notch signaling is associated with decreased VEGFR2 expression and EC proliferation in the anti-mesometrial decidua.

### 2.7. EC-Specific Loss of Jag1 Does Not Impact CD31^+^ EC Content in Pregnant Uteri at E7.5

To better understand the effect of loss of *Jag1* on decidual vasculature, flow cytometry was done to quantify the proportion of CD31^+^CD45^−^ ECs in the implantation site and myometrium of *Jag1**ΔEC* and control pregnancies at E7.5 (Figure 8A). Analysis of the uteri from nonpregnant C57BL/6 mice and pregnant C57BL/6 mice showed that an increase CD31^+^CD45^−^ EC content was the highest in the decidua of pregnant C57BL/6 mice at E7.5 (Figure 8B). Quantification of the percentages of CD31^+^CD45^−^ ECs in the decidua and myometrium of E7.5 *Jag1**ΔEC* and *Cdh5-Cre^ERT2^* control pregnancies revealed no significant differences (Figure 8C). Thus, despite loss of *Jag1* in ECs, the overall percentage of decidual CD31^+^ ECs is similar in *Jag1**ΔEC* and control pregnancies.

### 2.8. Vasculature Is Similar in Implantation Sites of Control and Jag1ΔEC Pregnancies

The recruitment and differentiation of mural cells, into pericytes and vascular smooth muscle cells, promotes vessel stabilization and is essential for vascular integrity [11,12,47,48]. We have shown that Jag1 is expressed in the ECs of SpAs at the mesometrial pole and in a subset of decidual mural cells (Figure 3A, [27]). To assess whether loss of *Jag1* in ECs impacts vascular density or mural cell content, implantation sites from *Jag1**ΔEC* mutant and *Cdh5-Cre^ERT2^* control pregnancies were stained for expression of CD31 and mural cell markers, PDGFRβ, NG2, and SMA (Appendix A).

Implantation sites stained for CD31 were quantified for percentage CD31^+^ ECs in the mesometrial pole and central region in the mesometrial region and anti-mesometrial region. In all regions, CD31^+^ expression is similar in *Jag1**ΔEC* mutant and control pregnancies, suggesting that loss of EC-specific *Jag1* does not impact decidual vascular density (Figure 9A–C). To assess whether loss of *Jag1* in ECs impacts mural cell content, implantation sites were analyzed for the expression of mural cell markers. No difference in the expression of NG2, PDGFRβ or SMA at the mesometrial pole of the mesometrial region (Figure 9D–F) was observed, suggesting that loss of endothelial *Jag1* does not impact mural cells in SpAs. Together, these data suggest that EC *Jag1* is not essential for decidual angiogenesis or maintenance of the SpA until E7.5.

## 3. Discussion

In this report, we focused on the role the endothelial Jag1 in decidual vasculature in the post-implantation, pre-placentation mouse uterus. When *Jag1* was deleted from the endothelium at E4.5 and pregnancies were assessed at E7.5, the start of placentation, we found that *Jag1∆EC* mutants maintain pregnancies with no change in embryo development. Loss of *Jag1* in SpA and capillary ECs resulted in a loss of Jag1 expression, but increased expression of Notch effectors, *Hey2* and *Nrarp,* consistent with a loss of Jag1 leading to increased Notch signaling activity. This correlated with an increase in Dll4 expression in capillary ECs, but not SpA ECs, in the mesometrium and in capillary ECs in the anti-mesometrium, suggesting that the increased Notch signaling activity was restricted to capillary ECs throughout the implantation site. In the anti-mesometrial decidua, increased expression of Notch1 and Notch signaling in angiogenic capillary ECs, leads to decreased endothelial VEGFR2 expression and EC proliferation. Despite aberrant Dll4 expression and Notch signaling activity, decidual vasculature development up to E7.5 in *Jag1∆EC* mutants was not altered in the time window we evaluated.

Angiogenesis occurs in the setting of both physiologic and pathologic growth and is largely controlled by pro-angiogenic signaling pathways, such as the VEGF and Notch pathways [16,19,49,50,51]. In the post-implantation, pre-placentation pregnant mouse uterus, physiologic angiogenesis is a dynamic process that involves sprouting and remodeling angiogenesis in the decidua that is regulated by VEGF/VEGFR2 [5,7,52] and Dll4/Notch signaling [53]. At E7.5, we found that the central region of the mesometrial decidua contains newer, immature, capillary networks with few, if any, associated mural cells, as there was little to no expression of NG2, PDGFRβ or SMA. In contrast, the ECs associated with NG2^+^, PDGFRβ^+^ pericytes in the anti-mesometrial region are consistent with the remodeling phase of angiogenesis [11,54]. Together, our findings agree with previous literature in suggesting that the central region and anti-mesometrial region are regions of early and active angiogenesis at E7.5.

Our evaluation of Notch proteins and ligands at E7.5 supports and extends our previous findings in the pre-placentation decidua [27,39]. We find Notch1 and Notch4 in the ECs of angiogenic capillaries in the central region, whereas, Notch1, Dll4 and Jag1, but not Notch4, are expressed in the ECs of remodeling capillaries in the anti-mesometrial region. Co-expression of Jag1 and Dll4 in sites of neovascularization and angiogenesis suggests that both these ligands could function in anti-mesometrial region ECs during decidual angiogenesis. In contrast, in the central region, Dll4 expression was detected in stromal, but not endothelial cells, whereas Jag1 expression was not detected in any central region cell-type. Further studies are needed to identify other Notch ligands, such as Jag2 or Dll1, that support angiogenesis and Notch signaling activity in the central region [27,39].

In mouse retinal vascular development [17], in skin wound healing [55] and in coronary plexus formation [56], EC-specific Jag1 has been shown to antagonize Dll4/Notch signaling. Disruption of endothelial *Jag1* leads to disinhibition of Dll4/Notch signaling, followed by increase in Dll4 expression and Notch activation, decreased *VEGFR2* and vascular density, and decreased EC proliferation [17,55]. Further, in wound angiogenesis, it has been proposed that a feedforward signaling mechanism maintains increased Dll4 expression [55]. We observed that loss of EC-specific *Jag1* increases Dll4 and Notch1 expression, as well as increases Notch signaling activity in areas of the decidua with remodeling angiogenesis. Consistent with increased Dll4/Notch signaling, we observed a decrease in VEGFR2 expression and EC proliferation, suggesting that Jag1 antagonizes Dll4/Notch signaling in angiogenic decidual capillaries. Our analysis does not distinguish the type of cellular interaction, *cis*- or *trans*-, that underlies Notch signaling activation in angiogenic decidual capillaries. Further studies are needed to determine if loss of a Jag1/Notch *trans*-interaction promotes Dll4/Notch *cis*-activation or *trans*-activation in anti-mesometrial decidual angiogenesis.

At E7.5, we did not observe a change in vascular density in the anti-mesometrial or central regions. Given that Jag1 is not expressed in central region ECs, the lack of change in Dll4 expression or vascular density in the central region, was not unexpected. In the anti-mesometrial region, where increased expression of EC Dll4 is observed, we predicted an increase in vascular density, as was previously reported with loss of EC-specific *Jag1* [17]. Had we found a change in vascular density, perhaps we would also have seen a change in the progression or survival of the pregnancy. Here, we show that the embryo can tolerate disturbances in the Jag1/Notch signaling by utilizing other members of the signaling pathway. Thus, while loss of Dll4/Notch signaling disrupts decidual angiogenesis [53], loss of endothelial Jag1, which leads to increased Dll4/Notch signaling, does not appear to be essential for early uterine angiogenesis prior to placentation.

SpAs in mice and humans are the distal branches of the uterine artery and are essential for bringing nutrients to the maternal fetal interface before development of the placenta. These arteries undergo important phases of circumferential remodeling, in part triggered by uNK cells, between E6.5 and E10.5 [3,9]. SpA remodeling involves the initial thickening of vascular smooth muscle lined vessel walls between E6.5 and E7.5 followed by their gradual thinning from E8.5 to E9.5. Trophoblast invasion of the SpAs mediates loss of smooth muscle coverage and the breakdown of extracellular matrix, resulting in thin walled vessels with wide lumens that allow for low resistance, high capacity maternal blood flow through the placenta to meet the demands of the growing embryo [3,9,57]. We show that at E7.5, the resident SpAs are still closely associated with mural cells that express NG2, PDGFRβ and SMA, which is consistent with mature vessels in the mesometrial pole that are not undergoing angiogenesis and have not, as yet, been remodeled. SpA ECs express Notch1 and Notch4 and Notch ligands, Jag1 and Dll4. In the pre-implantation uterus, we have previously shown that both Notch3 and Jag1 are expressed in mural cells in vessels in the stroma [27]. Whereas Jag1 is expressed in mural cells in vessels in the decidua, we did not find expression of Notch3 in post-implantation decidual vasculature at E6.5 [27] or E7.5 (data not shown). Thus, endothelial Jag1 and/or Dll4 could activate Notch1 and Notch4 in adjacent ECs to promote Notch signaling in SpA ECs, whereas SpA mural cells which do not express Notch proteins cannot be activated by endothelial Jag1 [39]. We found that loss of EC-specific *Jag1* does not result in increased Dll4 in SpAs in the mesometrial pole, suggesting no compensation or disinhibition of Dll4 with Jag1 loss, as was seen in the anti-mesometrial region capillaries. We also found that loss of EC-specific *Jag1* does not change vascular density or mural cell content in the SpAs. Together, our findings suggest Jag1/Notch signaling in SpA ECs is not essential for maintenance of decidual SpAs from implantation to E7.5. The impact of loss of EC-specific *Jag1* in SpAs may become evident later in gestation, nearing the completion of SpA transformation in the placenta.

Taken together, while we saw no vascular phenotype at E7.5 with loss of EC-specific *Jag1*, our data show that Jag1 and Dll4 mediate Notch signaling activity during decidual angiogenesis. Increased Dll4 in decidual capillary ECs is associated with increased Notch signaling, consistent with loss of inhibitory function of EC Jag1, which has been previously described with EC-specific loss of *Jag1* in retinal angiogenesis and wound healing [17,55]. Thus, our data identify another vascular bed, the newly formed uterine decidual capillaries, in which Jag1 functions as an antagonist of EC Dll4/Notch signaling during angiogenesis.

## 4. Materials and Methods

### 4.1. Animals

The Institutional Animal Care and Use Committee at Rutgers-New Jersey Medical School approved the studies (PROTO2018000086; October, 29, 2018). The experiments were designed to determine the role of endothelial Jag1/Notch signaling in the formation and function of uterine decidual blood vessels during early pregnancy development, prior to placentation. To delete Jag1 in the endothelium, *Cdh5-*Cre^ERT2^*,* an endothelial specific, tamoxifen inducible *Cre* transgenic strain (gift from Ralf Adams [38]) were crossed with *Jagged1^flox/flox^* mice [39]. To validate our protocol for tamoxifen administration with the *Cdh5-Cre^ERT2^* driver, tomato reporter mice *ROSA26 tdTomato* (B6.Cg-Gt(ROSA)26Sortm14(CAG-tdTomato)Hze/J, Jackson Laboratories) were used. *Cdh5-Cre^ERT2^* mice were bred to *Jagged1^flox/flox^* mice or *ROSA26 tdTomato* mice, thereby generating *Cdh5-Cre*;*Jagged1^fl/fl^* (*Jag1∆EC*) mutant females and *Cdh5-Cre^ERT2^*; *ROSA26 tdTomato* reporter females. *Cdh5-Cre^ERT2^* female mice were used as controls. All strains were maintained on a C57BL/6 background.

*Jag1∆EC* or control females, 8–12 weeks of age, were mated to C57BL/6 males for their first pregnancy. Identification of a vaginal plug in the morning was interpreted as successful mating. Noon was designated as embryonic day (E) 0.5. At E4.5, tamoxifen (0.1mg/g body weight (Sigma, Milwakee, WI, USA) was administered by oral gavage. Since the impact of loss of endothelial Jag1/Notch signaling on ovarian function and progesterone (P4) secretion is not known, P4 replacement was performed. Two extended-release P4 capsules (15 mg P4 per capsule, 21-d release, 4-mm diameter; Innovative Research of America, Sarasota, FL, USA) were placed subcutaneously. Pregnant female mice were euthanized at E7.5 and implantation sites without myometrium were isolated.

### 4.2. Quantitative Reverse Transcription-PCR

The myometrium was removed from implantation sites (IS) and total RNA was extracted from individual control and *Jag1**ΔEC* mutant IS using TRIzol (Invitrogen, Carlbad, CA, USA). Total RNA was treated with DNAse I (Invitrogen), reverse transcribed using qScript cDNA Supermix (Quanta Biosciences, VWR, Radnor, PA, USA) and gene expression were determined by quantitative (q) RT-PCR using the QuantiNova SYBR Green PCR Kit (Qiagen, Frederick, MD, USA) and the following primer sequences: *Jagged1* forward 5′-CTGCTTGAATGGGGGTCACT-3′, *Jagged1* reverse 5′-GCAGCTGTCAATCACTTCGC-3′; *Dll4* forward 5′-GTTGCCCTTCAATTTCACCT-3′, *Dll4* reverse 5′-AGCCTTGGATGATGATTTGG-3; *Hey2* forward 5′-AAGCGCCCTTGTGAGGAAAC-3′ *Hey2* reverse 5′-GGTAGTTGTCGGTGAATTGGAC-3′; *Nrarp* forward 5′-GCG TGG TTA TGG GAG AAA GAT-3′, *Nrarp* reverse 5′-GGG AGA GGA AAA GAG GAA TGA-3′. Relative expression levels were quantified using the 2^−ΔΔCt^ method and are expressed as fold change normalized to β-actin forward 5′-CGT GAA AAG ATG ACC CAG ATC-3′ and β-actin reverse 5′-CAC AGC CTG GAT GGC TAC GT-3′.

### 4.3. Analysis of Embryo Morphology

For histologic examination, implantation sites were fixed in Bouin’s solution (Sigma), paraffin embedded, section at 10 μm and stained with hematoxylin and eosin (H&E). For each implantation site, all sections having an implantation chamber were assessed by 4 different observers for the presence or absence of morphologic traits, including primitive streak, allantois, cranial neural fold and somites, that reflect the stage of embryonic development [44]. For each pregnant dam, two implantation sites were scored.

### 4.4. Immunofluorescence and Immunohistochemistry

Implantation sites with myometrium attached for immunofluorescence (IF) and immunohistochemistry (IHC) were fixed in 4% paraformaldehyde at 4 °C, infiltrated with 30% sucrose in PBS, embedded in Tissue-Tek^®^O.C.T™ Compound (Sakura Fine Technical Co. Ltd., VWR, Radnor, PA, USA), and cryosectioned at 7 μm. Interembryonic regions and central parts of the decidua were confirmed by H&E staining of every 5th section. For each implantation site, immunostaining was performed, as previously described [27,45] on at least 3 sections that were equally spaced along central parts of the decidua. The specificity of Notch protein and ligand primary antibodies was previously determined [27,58]. Slides stained with secondary antibody alone served as negative controls for IF and IHC staining. Antibodies are listed in the Appendix A. For colorimetric IHC, biotinylated anti-goat (Vector BA-5000, 1:400) or biotinylated anti-rat (Vector BA-4001, 1:200), the avidin/biotin blocking kit (Vector SP-2001), the Vectastain ABC kit and DAB substrate kit (Vector SK-4100) were used. For IF, Vectashield containing 40, 6-diamidino-2-phenylindole (DAPI) (Vector) was used for nuclear visualization and mounting. For each experiment, 2–3 implantation sites per pregnant dam were analyzed.

### 4.5. Flow Cytometry

Implantation sites were separated from myometrium and tissue finely minced in 100 uL of ice cold PBS. The finely minced tissue was digested in RPMI 1640 (Gibco) containing 1 mg/mL Collagenase A (Roche) and 0.1 mg/mL DNase I (Roche) for 30 min at 37 °C with horizontal shaking at 250 rpm. Mechanical dissociation by repeated passage through an 18 G needle was performed twice at 15 min intervals during enzymatic digestion. Ice cold RPMI containing 3% FBS (Atlanta Biologicals, R&D Systems, Minneapolis, MN, USA; RPMI-3)) was added and cells were filtered through a 100 μM nylon cell strainer.

Cells were stained in RPMI-3 for 30 min on ice in the dark with the following fluorochrome-conjugated anti-mouse antibodies: APC/Fire™ 750 anti-CD31 (BioLegend, clone MEC13.3, San Diego, CA, USA) and Super Bright 600 anti-CD45 (eBioscience, clone 30-F11, Carlsbad, CA, USA). Cells were washed twice and then resuspended in RPMI-3 containing 1μg/mL DAPI ((ThermoFisher, Carlsbad, CA, USA) and 2 mM EDTA (Corning, Carlsbad, CA, USA). Flow cytometry was performed using an Attune NxT (ThermoFisher, Carlsbad, CA, USA), LSRII (Becton, Dickinson and Company Sparks, MD, USA), or Fortessa (Becton, Dickinson and Company Sparks, MD, USA) flow cytometer, and analysis was performed using FlowJo™ for Mac v10.6.1 software (Becton, Dickinson and Company Sparks, MD, USA). Live endothelial cells were identified as DAPI^–^CD45^–^CD31^+^ cells.

### 4.6. Imaging

Images of H&E, IF and IHC samples were taken with the Keyence BZ-X710 All-in-One fluorescence microscope (Keyence, Osaka, Japan). Standard filters were used to image DAPI, Alexa Fluor 488 and Alexa Fluor 594. Images were taken using the 10×, 20×, and 40× objectives. Images were captured using Keyence BZ-X Viewer version 01.03 software.

### 4.7. Morphometric Analyses

Morphometric measurements of protein expression and blood vessel densities were determined on tissue immunostained for endothelial cell marker, CD31, mural cell markers NG2, PDGFRβ and SMA, and Notch ligands Jagged1 or Dll4. The mesometrial region and anti-mesometrial region of each implantation site were identified and further divided into mesometrial pole, defined as the area extending from the internal border of the myometrium to the uterine lumen, and the central region, defined as the areas of the implantation site lateral to and flanking the embryo and implantation chamber (Figure 1A,B). For blood vessel analysis, CD31 signal density was measured in 5 random 0.025 mm^2^ areas of the decidua in these three regions using ImageJ Software Version 2.0.0 (mesometrial pole, central region and anti-mesometrial region) [7]. Expression of Jag1 and Dll4 was determined by measurements of the signal densities and divided by CD31^+^ signal density or NG2+ or PDGFRβ^+^ signal density in 5 random 0.025 mm^2^ areas of the decidua. To quantify expression of N1ICD with respect to CD31, number of N1ICD^+^ nuclei was determined for 5 random 0.037 mm^2^ areas of anti-mesometrial decidua. The CD31 signal density was determined for each of these areas, and number of nuclei per square micrometer of CD31^+^ ECs was calculated. For analysis of VEGFR2 in CD31^+^ ECs, signal densities were evaluated in adjacent sections immunostained with VEGFR2 or CD31. The signal density was measured in 5 random 0.025mm^2^ areas of the decidua in like-regions for both VEGFR2 and CD31 and signal densities of VEGFR2 were divided by that of CD31^+^ cells. Sections from 2–3 implantation sites per each mouse were examined and were used for the analyses.

### 4.8. Statistics

Statistical analyses were performed with Prism version 8.4.1 (GraphPad, La Jolla, CA, USA). We analyzed these data to identify outliers and removed all outliers prior to performing statistical comparisons. Normality was determined using the Shapiro–Wilk test. Significant differences between medians were determined by unpaired Mann–Whitney U test. For normally distributed data, significant differences between means were determined by unpaired *t* test and ANOVA. Binary data were compared using Fisher’s exact test. Data are presented as median + IQR data or mean ± standard deviation (SD). Statistical significance was defined as *p* < 0.05.

## Figures and Tables

**Figure 1 ijms-21-06477-f001:**
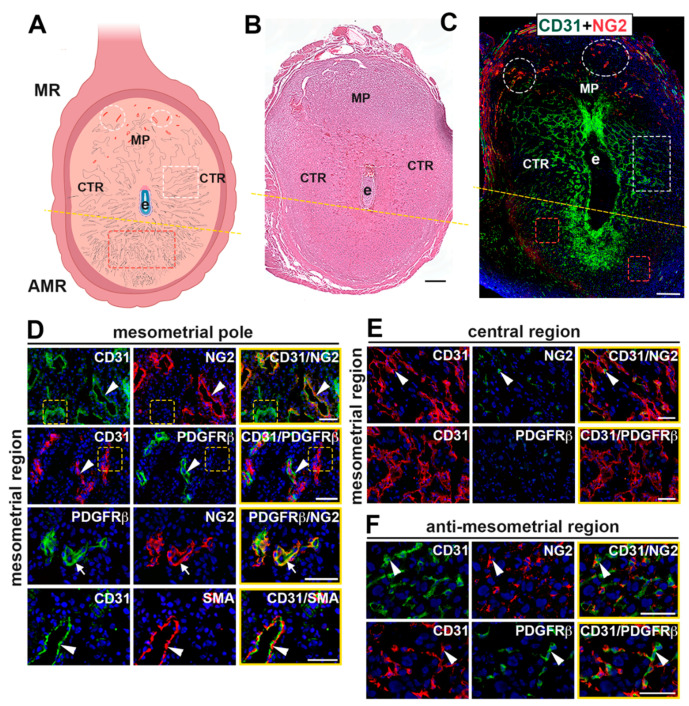
Characterization of the E7.5 uterine vasculature in wildtype mice. (**A**) Schematic representation of an E7.5 implantation site (created with BioRender.com). The implantation site is divided (represented by the dashed line) into two regions, the anti-mesometrial region (AMR) and the mesometrial region (MR), which is further divided into the mesometrial pole (MP) and central region (CTR). Labeled are spiral arteries (white ellipses) in the MP, capillaries (white rectangles) in the CTR, capillaries (red rectangles) in the AMR and the embryo (e). (**B**) H&E of an implantation site at E7.5 showing the embryo and central parts of the decidua. (**C**) An E7.5 implantation site double stained for the endothelial cell marker, CD31 and pericyte marker, NG2. Ellipses identify NG2^+^ SpAs at the MP. Rectangles identify CD31^+^ capillaries in the central region (white rectangle) and the AMR (red rectangle). (**D**–**F**) High magnification images of the vessels of the MR at the MP and CTR and AMR. Sections were double stained to detect expression of CD31 and mural cell markers, NG2, PDGFRβ or SMA. Merged images are outlined in yellow. CD31^+^ endothelial cells (ECs) are present throughout the implantation site in all three regions. (**D**) The vasculature in the MP is composed of SpAs seen as CD31^+^ ECs covered by NG2^+^, PDGFRβ^+^ and SMA^+^ mural cells (white arrowheads) and capillaries comprised of ECs that are not associated with NG2, PDGFRβ or SMA (yellow rectangles). NG2 and PDGFRβ are co-expressed (white arrow). (**E**) In the CTR, expression of NG2 is sparse and expression of PDGFRβ was not detected. Few CD31^+^ capillaries are associated with NG2^+^ mural cells (white arrowhead). (**F**) In the AMR, capillaries are comprised of CD31^+^ ECs closely associated with NG2^+^/PDGFRβ^+^ mural cells (white arrowheads). Scale bars = 200 μm in (**B**,**C**) and 50 μm in (**D**–**F**).

**Figure 2 ijms-21-06477-f002:**
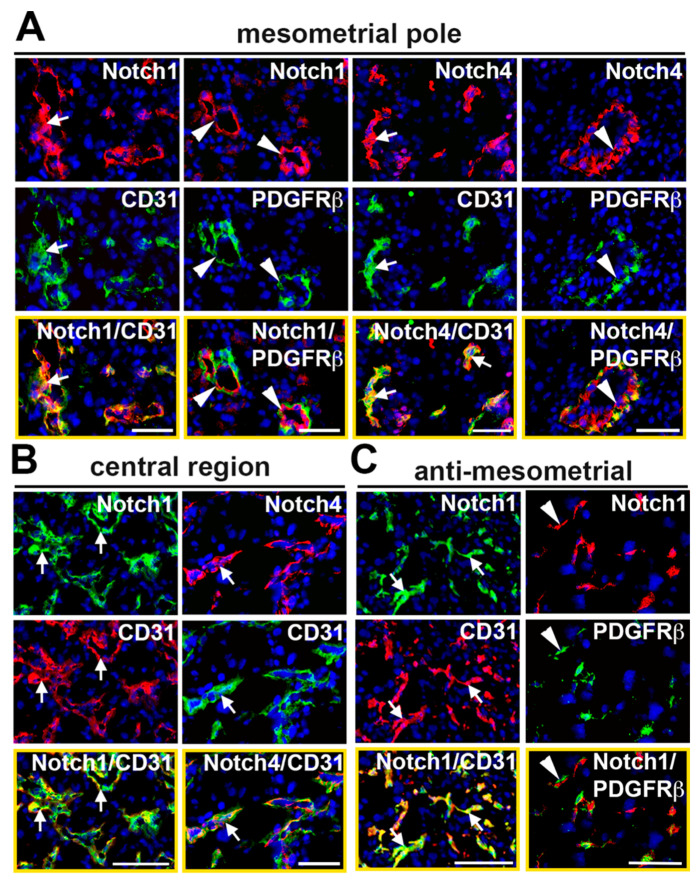
Expression of receptors, Notch1 and Notch4, in the E7.5 decidual vasculature. High magnification images of sections double stained to detect expression of Notch1 or Notch4 and the endothelial cell marker, CD31 or mural cell marker, PDGFRββ. Merged images are outlined in yellow. (**A**) In the MP, Notch1 and Notch4 are expressed in CD31^+^ SpA ECs (white arrows). PDGFRβ^+^ mural cells are associated with both Notch1^+^ and Notch4^+^ ECs in the SpAs (white arrowheads). (**B**) In the CTR, CD31^+^ capillary ECs express Notch1 and Notch4 (white arrows). (**C**) In the AMR, CD31^+^ capillary ECs express Notch1 (white arrows) and PDGFRβ^+^ pericytes are associated with Notch1^+^ cells (white arrowhead). AMR = anti-mesometrial; CTR = central region; MP = mesometrial pole. Scale bars = 50 μm.

**Figure 3 ijms-21-06477-f003:**
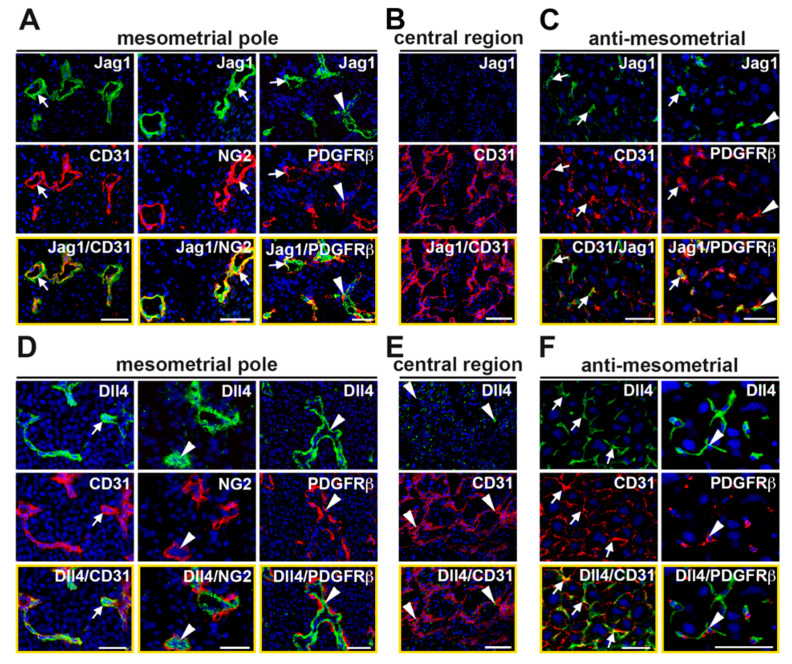
Expression of Notch ligands, Jag1 and Dll4, in the E7.5 decidual vasculature. High magnification images of sections double stained to detect expression of Jag1 or Dll4 and the endothelial cell marker, CD31 or mural cell marker, PDGFRβ. (**A**) In MP SpAs, Jag1 is expressed in CD31^+^ ECs (white arrow) and in NG2^+^ and PDGFRβ^+^ mural cells (white arrows). Some Jag1^+^ ECs are closely associated with PDGFRβ^+^/Jag1^−^ mural cells (white arrowheads). (**B**) Expression of Jag1 is not detected in the central region. (**C**) In the AMR, expression of Jag1 is sparse. Jag1 is expressed in CD31^+^ ECs and PDGFRβ^+^ mural cells (white arrows). Jag1^+^ cells are closely associated with PDGFRβ^+^ mural cells (white arrowheads). (**D**) In MP SpAs, Dll4 is expressed in CD31^+^ ECs (white arrows). NG2^+^ and PDGFRβ^+^ mural cells are closely associated with Dll4^+^ cells (white arrowheads). (**E**) In the CTR, Dll4 is expressed in a punctate pattern (white arrowheads). (**F**) In the AMR, Dll4 is expressed in CD31^+^ ECs (white arrows). Dll4^+^ cells are associated with PDGFRβ^+^ mural cells (white arowheads). AMR = anti-mesometrial; CTR = central region; MP = mesometrial pole. Scale bars = 50 μm.

**Figure 4 ijms-21-06477-f004:**
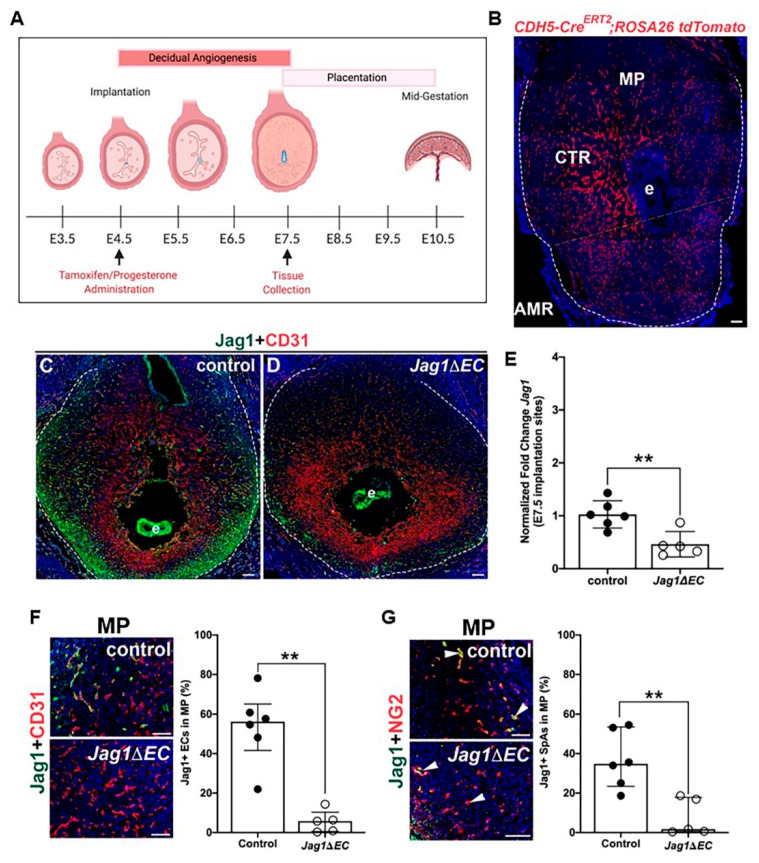
Tamoxifen-induced Cre recombination decreases expression of Jag1 in *Cdh5-Cre^ERT2^;Jag1^flox/flox^* (*Jag1ΔEC*) pregnancy. (**A**) Schematic of the mouse gestation timeline highlighting key processes involved, and experimental timeline (created with BioRender.com). At E4.5, tamoxifen is administered to induce Cre recombination and progesterone was administered to minimize secondary angiogenic defects in the uterus due to vascular defects in the ovaries. Pregnancies and uterine phenotype were assessed at E7.5. (**B**) Representative image of an implantation site at E7.5 from the *Cdh5-Cre^ERT2^*; *ROSA26 tdTomato* reporter. The decidua is within the dashed white. tdTomato expression, indicating Cre-induced recombination, is detected in a vascular pattern throughout the decidua. Dashed line denotes MR and AMR. (**C**,**D**) Representative images of implantation sites of *Cdh5-Cre^ERT2^* control and *Jag1ΔEC* pregnancies stained for Jag1 and CD31. Expression of Jag1 is reduced in the decidua (outlined with dashed white lines), but not the embryo of *Jag1ΔEC* mutants. (**E**) qRT-PCR determination of *Jag1* expression in implantation sites from control (*n* = 6) and *Jag1ΔEC* (*n* = 5) pregnancies. The relative expression level of *Jag1* was compared to β-actin. *Jag1* was significantly decreased in *Jag1ΔEC* mutants as compared to *Cdh5-Cre^ERT2^* controls. (**F**) Representative images of the MP of control and *Jag1ΔEC* implantation sites double stained for EC marker CD31 and Jag1. At the MP, expression of Jag1 in all CD31^+^ ECs is reduced in *Jag1ΔEC* mutants (*n* = 6) as compared to *Cdh5-Cre^ERT2^* controls (*n* = 5). (**G**) Representative images of the MP of control and *Jag1ΔEC* implantation sites double stained for Jag1 and NG2. Expression of Jag1 in the ECs of the NG2^+^ surrounded SpAs (white arrowheads) is reduced in *Jag1ΔEC* mutants (*n* = 5) as compared to *Cdh5-Cre^ERT2^* controls (*n* = 6). AMR= anti-mesometrial region; CTR = central region; e = embryo; MP = mesometrial pole. Scale bars = 100 μm in (**B**–**D**,**F**,**G**). Data shown as median + IQR; ** *p* < 0.01.

**Figure 5 ijms-21-06477-f005:**
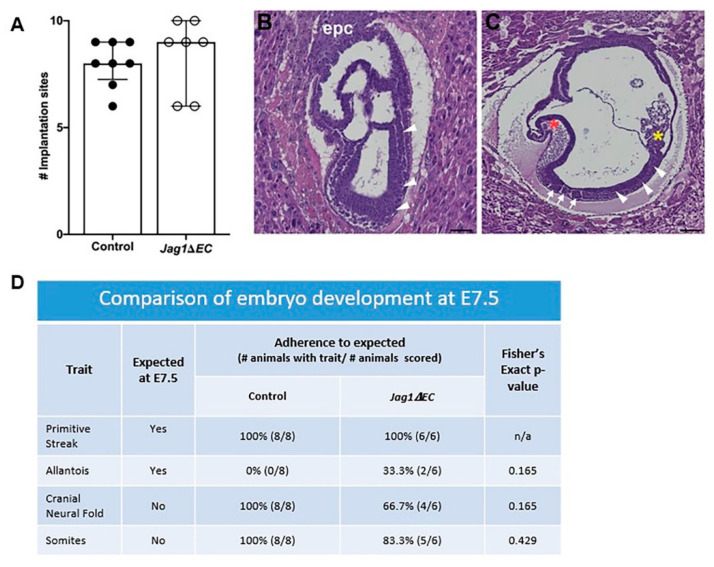
EC-specific loss of *Jag1* does not disrupt pregnancy progression or litter size at E7.5. (**A**) Number of implantation sites at E7.5 is similar between controls (*n* = 8) and *Jag1ΔEC* pregnancies (*n* = 7). (**B**,**C**) Representative H&E stained embryos, showing morphologic structures including the primitive streak (arrowheads), cranial neural fold (red asterisk), allantois (yellow asterisk) and somites (white arrows), observed in embryos at E7.5 (**B**) and E8.5 (**C**). (**D**) The presence of the primitive streak, allantois and cranial neural fold structures is similar in control and *Jag1ΔEC* pregnancies. Data were analyzed using Fisher’s exact test, comparing morphologic structures in control and *Jag1ΔEC* pregnancies to those expected in pregnancies at E7.5 [46].

**Figure 6 ijms-21-06477-f006:**
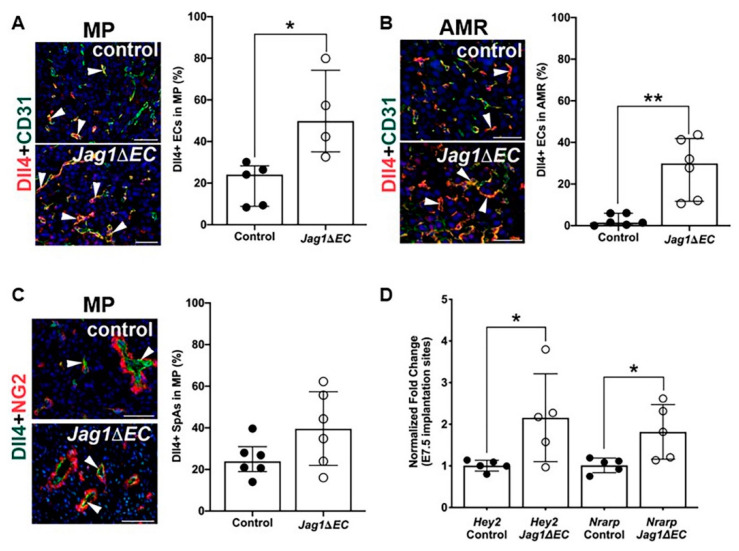
Expression of Notch ligand, Dll4, and Notch effectors are increased in ECs of *Jag1ΔEC* decidua. (**A**–**C**) Expression of Dll4 in capillaries and SpAs in *Cdh5-Cre^ERT2^* control and *Jag1∆EC* pregnancies was determined by double staining IF for Dll4 and CD31 or NG2. Representative images of the MP (**A**) and AMR (**B**) of control and *Jag1ΔEC* implantation sites. (**A**) Dll4 expression is increased in CD31^+^ ECs at the MP of *Jag1ΔEC* mutants (*n* = 4) as compared to *Cdh5-Cre^ERT2^* controls (*n* = 5). (**B**) Dll4 expression is increased in CD31^+^ capillary ECs in the AMR of *Jag1ΔEC* mutants (*n* = 6) as compared to *Cdh5-Cre^ERT2^* controls (*n* = 6). (**C**) Representative images of the MP of control and *Jag1ΔEC* implantation sites double stained for Dll4 and NG2. Expression of Dll4 in the ECs of the NG2^+^ surrounded SpAs is unchanged in *Jag1ΔEC* mutants (*n* = 6) as compared to *Cdh5-Cre^ERT2^* controls (*n* = 6). (**D**) qRT-PCR determination of Notch effector gene expression in implantation sites from control (*n* = 5) and *Jag1ΔEC* (*n* = 5) pregnancies. The relative expression level of each gene was compared to β-actin. *Nrarp* and *Hey2* are significantly increased in *Jag1ΔEC* mutants as compared to *Cdh5-Cre^ERT2^* controls. AMR = anti-mesometrial region; MP = mesometrial pole. Scale bars = 50 μm. Data shown are median +IQR. * *p* < 0.05, ** *p* < 0.01.

**Figure 7 ijms-21-06477-f007:**
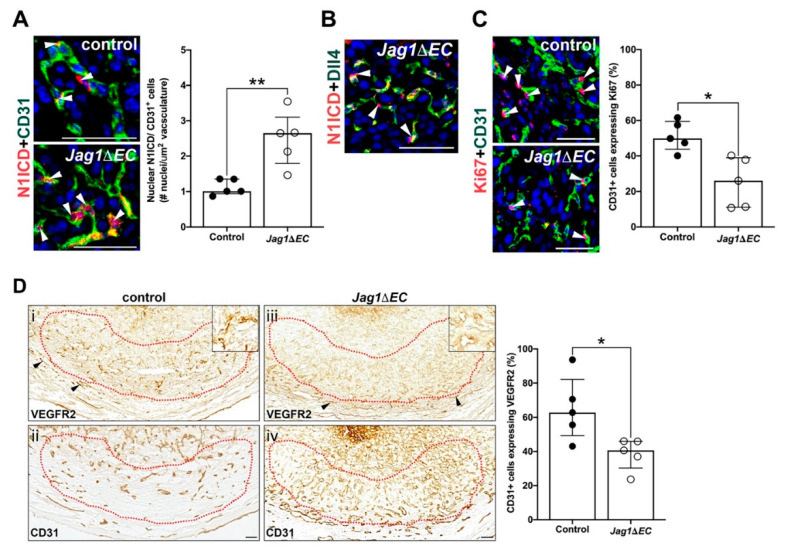
Increased Notch signaling in the anti-mesometrial decidua decreases EC proliferation and expression of VEGFR2. (**A**–**C**) High magnification images of sections double stained to detect expression of N1ICD and CD31, Dll4 or Ki67. (**A**) Expression of the N1ICD in CD31^+^ ECs in the AMR of *Cdh5-Cre^ERT2^* control and *Jag1∆EC* pregnancies was determined by double staining IF for N1ICD and CD31. Representative images of control and *Jag1ΔEC* implantation sites are shown. Nuclear expression of N1ICD (white arrowheads) is increased in CD31^+^ ECs in the AMR of *Jag1ΔEC* mutants (*n* = 5) as compared to *Cdh5-Cre^ERT2^* controls (*n* = 5). (**B**) N1ICD and Dll4 are expressed in adjacent cells in the AMR (white arrowheads) of *Jag1∆EC* pregnancies. (**C**) EC proliferation in the AMR of *Cdh5-Cre^ERT2^* control and *Jag1∆EC* pregnancies was determined by double staining IF for EC proliferation marker, Ki67 and CD31. Representative images of control and *Jag1ΔEC* implantation sites are shown. Ki67 expression (white arrowheads) is decreased in CD31^+^ ECs in the AMR of *Jag1ΔEC* mutants (*n* = 5) as compared to *Cdh5-Cre^ERT2^* controls (*n* = 5). (**D**) Expression of VEGFR2 with respect to CD31^+^ ECs in the AMR was determined by comparison of expression of VEGFR2 and CD31 in adjacent sections, in like-regions, of implantation sites from *Cdh5-Cre^ERT2^* control and *Jag1∆EC* pregnancies. Representative images of the AMR highlighting areas (decidua within dashed red lines) used to measure signal density from control and *Jag1ΔEC* implantation sites are shown (i–iv). VEGFR2 expression is decreased in CD31^+^ capillary ECs in the anti-mesometrial decidua of *Jag1ΔEC* mutants (*n* = 5) as compared to *Cdh5-Cre^ERT2^* controls (*n* = 5). Insets highlight VEGFR2 expression in decidual vessels. Black arrowheads highlight VEGFR2 expression in the myometrial vessels, which is similar in *Jag1ΔEC* mutants and controls. AMR = anti-mesometrial region; N1ICD = Notch1 intracellular domain. Scale bars = 50 μm. Data shown as median + IQR; * *p* < 0.05, ** *p* < 0.01.

**Figure 8 ijms-21-06477-f008:**
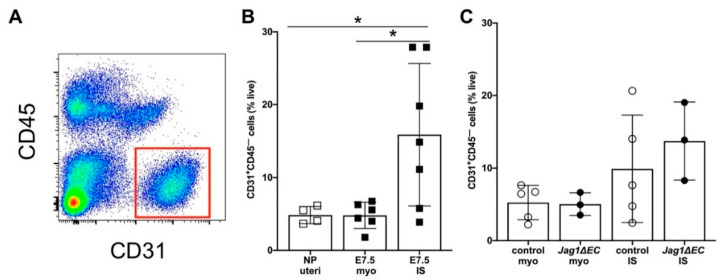
EC-specific loss of *Jag1* does not impact the frequency of CD31^+^ ECs in implantation sites. (**A**) Representative dot plot of flow cytometry (FCM) data showing gating strategy to identify CD31^+^ CD45^–^ EC populations in implantation sites from wild-type C57BL/6 mice at E7.5. (**B**) Results of FCM analysis to assess CD31^+^CD45^−^ cell populations in uteri from nonpregnant C57BL/6 mice, and in the myometrium and implantation sites from pregnant C57BL/6 mice at E7.5. The percentage of CD31^+^CD45^−^ ECs is significantly higher in implantation sites compared to the myometrium or nonpregnant uteri. (**C**) Results of FCM analysis to determine the percentage of CD31^+^CD45^−^ ECs in the myometrium and implantation sites of *Cdh5-Cre^ERT2^* control and *Jag1ΔEC* pregnancies at E7.5. The percentage of CD31^+^CD45^−^ ECs in the myometrium and implantation sites of *Cdh5-Cre^ERT2^* control and *Jag1ΔEC* pregnancies is similar. IS = implantation site; myo = myometrium; NP = nonpregnant; each group contains *n* = 3–5 individual mice. Data shown as median + IQR; * *p* < 0.05.

**Figure 9 ijms-21-06477-f009:**
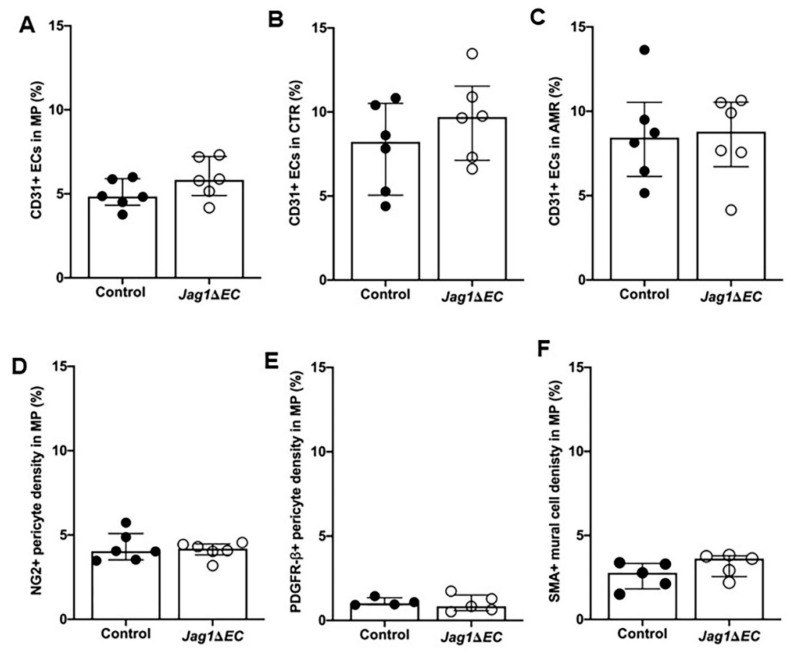
EC-specific loss of *Jag1* does not impact endothelial and mural cell density. (**A**–**C**) Expression of CD31^+^ was quantified to determine blood vessel density in each region of the implantation site. (**A**) Density of CD31^+^ ECs was similar in *Cdh5-Cre^ERT2^* control and *Jag1ΔEC* pregnancies in the MP (A) and CTR (**B**) of the MR and in the AMR (**C**). (**D**–**F**) Implantation sites stained for mural markers, NG2, PDGFRβ or SMA are assessed for mural cell content at the MP. Expression of NG2 (**D**), PDGFRβ (**E**) and SMA (**F**) is similar in *Cdh5-Cre^ERT2^* control and *Jag1**ΔEC* pregnancies in the MP. AMR = anti-mesometrial region; CTR = central region; MP = mesometrial pole; MR = mesometrial region. Data shown as median + IQR.

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
