# Peer review of "Endothelial Jagged1 Antagonizes Dll4/Notch Signaling in Decidual Angiogenesis during Early Mouse Pregnancy"

_ijms, 2020, doi:10.3390/ijms21186477_

Round 1
Reviewer 1 Report
This paper investigated the role of endothelial Jag1/Notch signaling in the formation and maintenance of the maternal decidual vasculature in early mouse pregnancy, using an inducible endothelial cell specific deletion of Jag1.
Immunohistochemistry was first utilised to characterize the decidual vasculature and the expression of members of the Notch signalling pathway. The Jag1ΔEC model was then validated, demonstrating significant decreases in endothelial Jag1 expression.
The impact of the deletion was then analysed in several ways; including pregnancy and embryo development, downstream effector expression, EC and mural cell content.
The study reveals that endothelial Jag1 is not required for decidual angiogenesis or spiral artery maintenance.
This is an interesting paper and I have very few concerns.
This paper is clearly written and well presented, although the frequent use of abbreviations is sometimes confusing and detracts from the flow of the writing.
The model used is interesting and the authors have addressed their question in a comprehensive way.
The figures are appropriate, if sometimes a little small (e.g. thumbnails Fig 2).
The PDF has mistranslated numerous symbols.
The bibliography needs editing for consistent format (e.g. use of full stops; use of short and long Journal names).
This reviewer could not access the supplemental files.
Reviewer 2 Report
This is a strong developmental biology manuscript investigating the role of endothelial Jagged1 in decidual angiogenesis during early mouse pregnancy. The study is well conceived, thoroughly done, and provides important but negative data demonstrating that Jag1 is not essential for early uterine angiogenesis. The claim that Jag1 functions in decidual capillaries as an antagonist of endothelial Dll4/Notch signaling is underdeveloped. This area should be further complemented by additional quantitative measures and experiments or address alternative explanations in the discussion and remove the firm conclusion from the abstract.
Mechanisms of Notch pathway trans-activation and cis-inhibition need to be considered. Do ligand and receptor expression overlap or are ligand and receptor expressed in adjacent cells in specific endothelial cell populations? How might this impact trans-activation and cis-inhibition? Are Jag1 and Dll4 expressed in the same cell in this context?
To firm up conclusion that Dll4/Notch signaling is increased, a comparison of Dll4 overlap and adjacent cellular expression with receptor expressing cells is needed. Is Notch receptor expression increased (level and cell number)?
Is there increased Notch signaling or more Hey2 and Nrarp positive cell fates acquired? Accompanied anti-Notch1 V1744 immunostaining would verify increased Notch signaling.
Minor concerns:
Efficiency of recombination is not quantified.
Is Jag1 presence in MP ECs and SpAs on par with percentage of reporter expression in these populations?
Quantitative size parameters, such as somite number are not reported as part of the phenotypic analysis. Is there a slight developmental delay?
Round 2
Reviewer 2 Report
The authors have added important and clear data to improve the manuscript, but caution to use the correct terminology is important. The D1E11 XP; Catalog #3609S, Cell Signaling antibody is against the cytoplasmic domain of Notch1. The antibody detects full-length Notch1, membrane tethered Notch1, and the activated form of Notch1 (referred to N1ICD in the Notch field). The presence of cells that are positive for Notch1 detected by anti-D1E11 does not automatically mean they have or are experiencing Notch1 activation. Only when using the V1744, #4147S, Cell Signaling antibody can this be claimed. In Figure 7 quantification, do the authors only count cells that have nuclear localization detected by D1E11? The second concern is the interpretation that an increase of Dll4 and “N1ICD” in the same cell indicates increased Notch signaling. Canonical Notch signaling is elicited by ligands expressed on cells adjacent to the Notch receptor (trans-activation). Ligands expressed within the same cell as the receptor is known to inhibit Notch activation (cis-inhibition). Therefore, the previous considerations remain to be discussed - mechanisms of Notch pathway trans-activation and cis-inhibition in this context. Do ligand and receptor expression overlap or are ligand and receptor expressed in adjacent cells in specific endothelial cell populations? How might this impact trans-activation and cis-inhibition? Are Jag1 and Dll4 expressed in the same cell type in this context?
